# How to Lose Inherent Counterfactuality in Reinforcement Learning

**Ezgi Korkmaz**

## Abstract

Learning in high-dimensional MDPs with complex state dynamics became possible with the progress achieved in reinforcement learning research. At the same time, deep neural policies have been observed to be highly unstable with respect to the minor variations in their state space, causing volatile and unpredictable behaviour. To alleviate these volatilities, a line of work suggested techniques to cope with this problem via explicitly regularizing the temporal difference loss to ensure local $\epsilon$-invariance in the state space. In this paper, we provide theoretical foundations on the impact of robust, i.e. adversarial, training on reinforcement learning. Our comprehensive theoretical and experimental analysis reveals that standard reinforcement learning inherently learns counterfactual values while recent training techniques that focus on explicitly enforcing $\epsilon$-local invariance cause policies to lose counterfactuality, and further result in learning misaligned and inconsistent values. In connection to this analysis, we further highlight that this line of training methods breaks the core intuition and the true biological inspiration of reinforcement learning, sacrifices essential inherent skills that enable reasoning and generalization, and introduces an intrinsic gap between how natural intelligence understands and interacts with an environment in contrast to AI agents trained via $\epsilon$-local invariance methods. The misalignment, inaccuracy and the loss of counterfactuality revealed in our paper further demonstrate the need to rethink the approach in establishing truly reliable and generalizable reinforcement learning policies.

## 1 Introduction

Inspired by the learning dynamics and cognitive abilities of natural intelligence (Watkins, 1989; Sutton, 1984; Kehoe et al., 1987; Romo & Schultz, 1990; Montague et al., 1996; Schmidhuber, 1999; Schultz et al., 1993; Pan et al., 2005), reinforcement learning research has been the focal point of immense research progress (Mnih et al., 2015; Hasselt et al., 2016; Korkmaz, 2026). Deep reinforcement learning has become an emerging field in the past decade with the introduction of deep neural networks as function approximators leading to learning policies that can surpass human cognitive abilities in highly complicated tasks by solely interacting with a given environment through trial and error without any supervision, consequently resulted in building AI agents that can reason and strategize (Mnih et al., 2015; Kapturowski et al., 2023; Krishnamurthy et al., 2024; Korkmaz, 2025). In parallel, advances in neuroscience revealed the precise structures and neural circuitry dedicated to the computation of counterfactual state-action values in the human brain, and how these values are later compared to make decisions. In the specialized neural circuitry that underpins decision-making a compelling functional divide has been identified: while the prefrontal cortex encodes the expected values of executed actions, the dorsomedial frontal cortex plays a critical role in the analysis of counterfactual decisions providing the mechanisms for learning that can reason and generalize (Wunderlich et al., 2009; Lau & Glimcher, 2007; Klein-Flügge et al., 2016).

Beyond the initial neuroscientific inspiration, reinforcement learning further offers strong, mathematically provable, asymptotic guarantees on its ability to learn policies for solving complex problems via trial and error (Sutton, 1984; Watkins & Dayan, 1992). Nonetheless, a recent body of work exposed critical safety concerns of reinforcement learning (Korkmaz, 2022; 2024), and consequently, a new class of algorithms has emerged that modify standard reinforcement learning algorithms to ensure reliability and safety in deep reinforcement learning (Madry et al., 2018; Huan et al., 2020; Li et al., 2024).

In this paper, we investigate the core principles of reinforcement learning, and we analyze the theoretical underpinnings of counterfactuality and alignment in connection to the neuroscientific analysis of natural intelligence, and the consequential effects of trying to ensure safety in reinforcement learning. Our analysis discovers that the line of research focused on safety fails to deliver the guaranteed outcomes, and further risks potentially significant changes to the behavior and semantics of the trained policies that disrupts the foundations of reinforcement learning and its inherent capabilities. Essentially in this paper we aim to seek answers for the following questions: *(i) What are the consequences of efforts to explicitly impose safety on reinforcement learning? (ii) What are the underlying reasons for preserving the core intuitive principles, neuroscientific foundations, and inherent capabilities of reinforcement learning?* To be able to answer these questions we focus on the foundations of reinforcement learning and its alignment with natural intelligence, and make the following contributions:

**Contributions.** We first provide a theoretically well-founded rigorous analysis of the state-action value function learnt by methods explicitly enforcing $\epsilon$-local invariance and standard reinforcement learning in Section 3. Our analysis uncovers fundamental insights into how $\epsilon$-local invariance imposition alters the very fabric of an agent's learned value judgments. Our paper is the first one that demonstrates, both theoretically and empirically, that methods explicitly enforcing robustness in fact fundamentally disrupt the inherent learning processes of standard reinforcement learning, consequentially leading to the subversion and loss of essential skills. Our analysis reveals that reinforcement learning possesses an inherent ability for counterfactual reasoning and is naturally aligned with human decision-making processes, while a recent line of work focusing on enforcing reinforcement learning to be explicitly robust causes RL policies to lose the inherent counterfactual ability and results in learning policies that are inaccurate, inconsistent and misaligned. We then conduct experiments in MDPs with high-dimensional state spaces from the Arcade Learning Environment (ALE) in Section 4, and our comprehensive study verifies the theoretical analysis and demonstrates a critical trade-off. Our findings reveal that reinforcement learning naturally retains core skills that align with the value assignment of natural intelligence which allows them to reason and generalize. However, subjecting them to robust training shatters this elegant relationship and eradicates this intrinsic counterfactual ability and alignment. Our paper establishes the foundational principle of an intrinsic trade-off between counterfactuality and robustness and further uncovers the core mechanisms driving this fundamental trade-off as a direct result of explicit imposition of safety, which fundamentally sacrifices the core attributes allowing reasoning and generalization.

## 2 BACKGROUND AND PRELIMINARIES

**Markov Decision Process.** An MDP is represented by a tuple $\mathcal{M} = (S, \mathcal{A}, P, r, \rho_0, \gamma)$ where $S$ is a set of continuous states, $\mathcal{A}$ is a discrete set of actions, $P$ is a transition probability distribution on $S \times \mathcal{A} \times S$, $r : S \times \mathcal{A} \rightarrow \mathbb{R}$ is a reward function, $\rho_0$ is the initial state distribution, and $\gamma$ is the discount factor. The objective in reinforcement learning is to learn a policy $\pi : S \rightarrow \Delta(\mathcal{A})$ which maps states to probability distributions on actions in order to maximize the expected cumulative reward $R = \mathbb{E} \sum_{t=0}^{T-1} \gamma^t r(s_t, a_t)$ where $a_t \sim \pi(s_t)$. In $\mathcal{Q}$-learning (Watkins, 1989) the goal is to learn the optimal state-action value function $\mathcal{Q}^*(s, a) = R(s, a) + \gamma \sum_{s' \in S} P(s'|s, a) \max_{\hat{a} \in \mathcal{A}} \mathcal{Q}^*(s', \hat{a})$. Thus, the optimal policy is determined by choosing the action $a^*(s) = \arg\max_a \mathcal{Q}^*(s, a)$ in state $s$.

**Adversarial Crafting and Training.** Concerns regarding $\epsilon$-invariance start with the work of Goodfellow et al. (2015), who observed that perturbations that are imperceptible to natural intelligence can in fact change the decision of a deep neural network and further suggested a fast method to produce such perturbations based on the linearization of the cost function used in training the network. Kurakin et al. (2016) proposed the iterative version of the fast gradient sign method proposed by Goodfellow et al. (2015) inside an $\epsilon$-ball

$$x_{\text{adv}}^{N+1} = \text{clip}_\epsilon(x_{\text{adv}}^N + \alpha \text{sign}(\nabla_x J(x_{\text{adv}}^N, y))) \tag{1}$$

in which $J(x, y)$ represents the cost function used to train the deep neural network, $x$ represents the input, and $y$ represents the output labels. While several other methods have been proposed Korkmaz (2024) using a momentum-based extension of the iterative fast gradient sign method,

$$v_{t+1} = \mu \cdot v_t + \frac{\nabla_{s_{\text{adv}}} J(s_{\text{adv}}^t + \mu \cdot v_t, a)}{\|\nabla_{s_{\text{adv}}} J(s_{\text{adv}}^t + \mu \cdot v_t, a)\|_1} \quad , \quad s_{\text{adv}}^{t+1} = s_{\text{adv}}^t + \alpha \cdot \frac{v_{t+1}}{\|v_{t+1}\|_2}$$

robust training, i.e. $\epsilon$-invariance training, has mostly been conducted with perturbations computed by projected gradient descent, i.e. PGD, proposed by Madry et al. (2018) (i.e. Equation 1).

**Neuroscientific Results and Alignment with Natural Intelligence Decisions Making.** A notable aspect of human cognitive decision-making is the assignment of counterfactual values to alternative, unchosen decisions (Wunderlich et al., 2009; Lee et al., 2012; Phillips et al., 2019). This mechanism serves to inform future decision-making by preserving a clear ordering of both factual and counterfactual outcomes and is a key attribute of the decision-making process that enables generalization and reasoning (Hoeck et al., 2015; Phillips et al., 2019; Grabenhorst & Rolls, 2011). Notably, the results in Figure 1 report analysis of fMRI scans of human brains during a decision-making task to identify a neural structure that compares the values of chosen and unchosen options for a particular decision. The results demonstrate that the value of each option was encoded in this structure, and that the actual decisions made were accurately predicted by these values (Klein-Flügge et al., 2016).

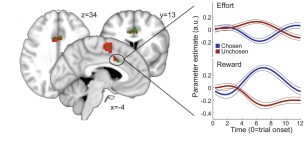

Figure 1: Human decision making and value assignment for options (Klein-Flügge et al., 2016).

**Concerns on Reliability of Deep Neural Policies.** The initial investigation on volatilities of deep neural policies was conducted based on testing $\epsilon$-invariance via the utilization of the fast gradient sign method proposed by Goodfellow et al. (2015). Robustness and reliability of reinforcement learning policies has been further analyzed, and some studies argued the existence of shared $\epsilon$-variant directions across MDPs and algorithms can be due to an underlying linear structure learnt by policies (Korkmaz, 2022). While several studies focused on improving optimization techniques for computing optimal perturbations, a line of research focused on making deep neural policies resilient to these perturbations. In particular, Pinto et al. (2017) proposed to model the dynamics between the adversary and the deep neural policy as a zero-sum game where the goal of the adversary is to minimize expected cumulative rewards of the deep neural policy. Gleave et al. (2020) approached this problem with an adversary model which is restricted to take natural actions in the MDP instead of modifying the observations with $\ell_p$-norm bounded perturbations. The authors model this dynamic as a zero-sum Markov game and solve it via self-play. Recently, Huan et al. (2020) proposed to model this interaction as a state-adversarial MDP, and further claimed that their proposed algorithm SA-Double Deep Q-Network (SA-DDQN) learns theoretically certified robust $\epsilon$-invariant policies against both natural noise and perturbations. Recent studies surprisingly revealed that robust, i.e. adversarial, training exhibits generalization issues with unpredictable behaviour and larger oscillations compared to standard reinforcement learning (Korkmaz, 2023; 2024). Despite these observations on generalization, currently still a large body of work is produced on explicitly optimizing variants of $\epsilon$-invariance training, without any foundational analysis and explanation on how precisely this class of algorithms affect the inherent abilities of standard reinforcement learning and why this line of approach might not be the way of achieving true safety and reliability.

# 3 THE CORE INTUITION OF REINFORCEMENT LEARNING: THE INHERENT COUNTERFACTUALITY

Our extensive analysis and results discover that $\epsilon$-invariance training methods break the core intuitive principles of reinforcement learning and erode the inherent skills of RL policies up to the level of learning random values for counterfactual actions. Our analysis reveals that a key and strong attribute of reinforcement learning that allows generalization and reasoning is lost when subjected to robust, i.e. adversarial, training. Our study provides evidence that the application of $\epsilon$-invariance training fails to fully address the critical issues of robustness and safety in modern AI. The observed misalignment between this training methodology and desirable system behavior reveals a key dichotomy: while certified training aims for provable guarantees, it appears to diverge from the inherent principles that underpin the core attributes of natural intelligence with regard to reasoning and generalization.

The theoretically motivated $\epsilon$-locally invariant ($\epsilon$-LI) training techniques achieve certified defense against perturbations inside the $\epsilon$-ball, $\mathcal{D}_\epsilon(s) = \{\hat{s} : \|s - \hat{s}\|_p \le \epsilon\}$, forming the current foundations of the robust reinforcement learning. However, we provide foundational evidence that this approach induces significant changes in the $\mathcal{Q}$-function where the state-action value function no longer accurately represents the MDP. In particular, $\epsilon$-locally invariant ($\epsilon$-LI) training causes deep neural policies to learn misaligned, inaccurate, overestimated state-action value functions while causing standard

reinforcement learning to lose it's inherent counterfactuality. Furthermore, we connect and highlight the neural processing of decision making of natural intelligence, core intuition of reinforcement learning and certified training (Wunderlich et al., 2009; Lau & Glimcher, 2007; Grabenhorst & Rolls, 2011). Our results reveal that certified training disrupts the core intuition of reinforcement learning and leads to learning policies that are disjoint and orthogonal to natural intelligence decision making, i.e. the true biological inspiration of reinforcement learning (Romo & Schultz, 1990; Montague et al., 1996). In the remainder of this section we will provide the theoretical foundations on: *i. Why we need to preserve the core intuition of reinforcement learning*, and *ii. What factors precisely disrupts the essential core skills learnt by reinforcement learning*. The theoretical underpinning of $\epsilon$-invariance training methods is derived from Danskin's theorem.

**Theorem 3.1** (Danskin (1967)). *Let $\mathcal{X}$ be a compact topological space $f : \mathbb{R}^n \times \mathcal{X} \to \mathbb{R}$, $f(\cdot, x)$ is differentiable for every $x \in \mathcal{X}$, $x^*(\theta) = \{x \in \arg\max_{x \in \mathcal{X}} f(\theta, x)\}$ and $\nabla_\theta f(\theta, x)$ is continuous on $\mathbb{R}^n \times \mathcal{X}$. Then the max function $\kappa(\theta) = \max_{x \in \mathcal{X}} f(\theta, x)$ is locally Lipschitz continuous, directionally differentiable, and its directional derivatives satisfy $\kappa'(\theta, h) = \sup_{x \in x^*(\theta)} h^\top \nabla_\theta f(x, \theta)$. Furthermore, if the set $x^*(\theta)$ has size one i.e. there is a unique maximizer $x_\theta^*$ then $\nabla_\theta \kappa(\theta) = \nabla_\theta f(\theta, x_\theta^*)$.*

Danskin's theorem provides a method to compute the gradient of a function that is defined in terms of a maximization over a set. With this theoretically well-motivated start, a line of algorithms have been proposed to make models reliable. The approach of $\epsilon$-invariance training techniques is based on editing the standard $\mathcal{Q}$-learning update. This change made to the update is designed to penalize $\mathcal{Q}$-functions for which a perturbed state $\hat{s} \in \mathcal{D}_\epsilon(s)$ can change the identity of the highest $\mathcal{Q}$-value action. Formally, the canonical definition used in the literature is

**Definition 3.2** (*Robust reinforcement learning*). Within an $\epsilon$-ball $\mathcal{D}_\epsilon(s) = \{\hat{s} : \|s - \hat{s}\|_p \leq \epsilon\}$ the reinforcement learning policy is robust, i.e. the $\mathcal{Q}$ function is $\epsilon$-locally invariant if $\forall s \in \mathcal{S}, \hat{s} \in \mathcal{D}_\epsilon(s)$, $\arg\max_a \mathcal{Q}(s, a) = \arg\max_a \mathcal{Q}(\hat{s}, a)$.

Now we will prove that there is a fundamental trade-off between accurate estimation of $\mathcal{Q}$-values and robustness. In particular, the optimal state-action value function $\mathcal{Q}^*$ is not $\epsilon$-invariant, but there is a $\epsilon$-invariant state-action value function $\mathcal{Q}_\theta$ that overestimates the optimal state-action values.

**Theorem 3.3** (*Inherent trade-off between estimation and robustness*). *Let $\epsilon > 0$. In the linear function approximation setting, there is an MDP such that all linear-state action value functions matching the optimal state-action values $\mathcal{Q}^*$ are not $\epsilon$-invariant. Furthermore, there is a linear state-action value function $\mathcal{Q}_\theta$ that is $\epsilon$-invariant, but overestimates the optimal state-action values while maintaining the correct optimal action.*

*Proof.* Let there be two states $s_1$ and $s_2$ such that $\|s_1 - s_2\|_2 = 1$. Further suppose that the optimal state-action values satisfy $\mathcal{Q}^*(s_1, a_1) = \epsilon/10$, $\mathcal{Q}^*(s_1, a_2) = 0$, $\mathcal{Q}^*(s_2, a_1) = 0.8$, and $\mathcal{Q}^*(s_2, a_2) = 1.0$. Next let $\mathcal{Q}_\theta(s, a)$ be any linearly parameterized state-action value function that agrees with $\mathcal{Q}^*(s, a)$ on the states $s_1$ and $s_2$. Consider the one-dimensional functions $\Psi_1(\xi) = \mathcal{Q}_\theta((1 - \xi) \cdot s_1 + \xi \cdot s_2, a_1)$ and $\Psi_2(\xi) = \mathcal{Q}_\theta((1 - \xi) \cdot s_1 + \xi \cdot s_2, a_2)$ which are the restriction of $\mathcal{Q}_\theta(s, a)$ to the line segment from $s_1$ to $s_2$. By linearity of $\mathcal{Q}_\theta$ we also have that both $\Psi_1$ and $\Psi_2$ are linear. Furthermore, since $\mathcal{Q}_\theta$ agrees with $\mathcal{Q}^*$ at $s_1$ and $s_2$, we know the values of both functions at two points i.e. $\Psi_1(0) = \mathcal{Q}^*(s_1, a_1)$, $\Psi_1(1) = \mathcal{Q}^*(s_2, a_1)$, $\Psi_2(0) = \mathcal{Q}^*(s_1, a_2)$, and $\Psi_2(1) = \mathcal{Q}^*(s_2, a_2)$. As $\Psi_1$ and $\Psi_2$ are linear functions on $\mathbb{R}$, the values at two points are sufficient to uniquely determine the functions. In particular we have

$$\Psi_1(\xi) = (0.8 - \epsilon/10)\xi + \epsilon/10 \quad \text{and} \quad \Psi_2(\xi) = \xi$$

Note that these two lines intersect at the point $\hat{\xi} = \frac{\epsilon}{2 + \epsilon}$. Let $\hat{s} = (1 - \hat{\xi}) \cdot s_1 + \hat{\xi} \cdot s_2$. Since the lines of $\Psi_1$ and $\Psi_2$ intersect at $\hat{\xi}$, we conclude that $\mathcal{Q}_\theta(\hat{s}, a_2) \geq \mathcal{Q}_\theta(\hat{s}, a_1)$. However, $\mathcal{Q}_\theta(s_1, a_1) > \mathcal{Q}_\theta(s_1, a_2)$. Furthermore, $\|s_1 - \hat{s}\| = \frac{\epsilon}{2 + \epsilon} < \epsilon$. Thus, $\mathcal{Q}_\theta$ is not $\epsilon$-invariant. However, if we instead choose new parameters $\theta'$ for the state-action value function so that $\mathcal{Q}_{\theta'}(s_1, a_1) = 0.8$ and $\mathcal{Q}_{\theta'}(s_1, a_2) = 0.7$ one can easily check that $\mathcal{Q}_{\theta'}$ is $\epsilon$-invariant for all $\epsilon < 0.1$. Furthermore, observe that $\mathcal{Q}_{\theta'}$ gives the correct ranking of actions in state $s_1$, but overestimates the optimal state-action value by a factor of $8/\epsilon$. $\square$

The results reported in Section 4 verify the theoretical analysis on the fundamental trade-off between estimation and $\epsilon$-invariance in neural-network approximation of the $\mathcal{Q}$-function. Now we will further theoretically analyze the effects of canonical $\epsilon$-invariance training techniques (Huan et al., 2020).

**Definition 3.4** (*Baseline Certified $\epsilon$-invariance Training*). The regularizer that achieves certified robustness, i.e. $\epsilon$-invariance, inside the $\epsilon$-ball $\mathcal{D}_\epsilon(s) = \{\hat{s} : \|s - \hat{s}\|_\infty \leq \epsilon\}$ for $\mathcal{Q}_\theta(s, a)$ is

$$\mathcal{R}(\theta) = \sum_s (\max_{\hat{s} \in \mathcal{D}_\epsilon(s)} \max_{a \neq \arg\max_a \mathcal{Q}_\theta(s,a)} \mathcal{Q}_\theta(\hat{s}, a) - \mathcal{Q}_\theta(\hat{s}, \arg\max_a \mathcal{Q}_\theta(s, a))).$$

The certified training algorithm proceeds by adding $\mathcal{R}(\theta)$ to the standard temporal difference loss $\mathcal{L}_\mathcal{H}(r(s, a) + \gamma \max_{a'} \mathcal{Q}^{\text{target}}(s', a') - \mathcal{Q}_\theta(s, a)) + \mathcal{R}(\theta)$.

Now we will show that changing the standard $\mathcal{Q}$-update will cause losing counterfactuality $\forall a \in \mathcal{A}_s^\perp$ where $\mathcal{A}_s^\perp := \{a | a \neq \arg\max_{\hat{a}} \mathcal{Q}(s, \hat{a})\}$, and overestimation of the state-action values. For this now let us look at the MDP $\mathcal{M}$ where two states parametrized by feature vectors $s_1, s_2 \in \mathbb{R}^n$, with three possible actions $\{a_i\}_{i=1}^3$ in each state where taking any of the actions in state $s_1$ leading to a transition to state $s_2$ and vice versa. Let $1 > \gamma > 0$ be the discount factor, and let $\delta > \eta > 0$ be small constants with $\gamma > \delta$. The rewards for each action are as follows: $r(s_1, a_1) = 1 - \gamma$, $r(s_1, a_2) = \eta - \gamma$, $r(s_1, a_3) = \delta - \gamma$, $r(s_2, a_1) = \eta - \gamma$, $r(s_2, a_2) = 1 - \gamma$, and $r(s_2, a_3) = \delta - \gamma$. Clearly, the optimal policy is to always take action $a_1$ in state $s_1$, and action $a_2$ in state $s_2$ as these are the only actions giving positive reward. Thus the optimal state-action values are given by: $\mathcal{Q}^*(s_1, a_1) = \mathcal{Q}^*(s_2, a_2) = \sum_{t=0}^\infty (1 - \gamma)\gamma^t = 1$, $\mathcal{Q}^*(s_1, a_2) = \mathcal{Q}^*(s_2, a_1) = \eta - \gamma + \gamma \sum_{t=0}^\infty (1 - \gamma)\gamma^t = \eta$ , and $\mathcal{Q}^*(s_1, a_3) = \mathcal{Q}^*(s_2, a_3) = \delta - \gamma + \gamma \sum_{t=0}^\infty (1 - \gamma)\gamma^t = \delta$. Let the $\mathcal{Q}$-function be linearly parametrized by $\theta = (\theta_1, \theta_2, \theta_3)$ so that $\mathcal{Q}_\theta(s, a_i) = \langle \theta_i, s \rangle$. Finally, let $\Phi_i$ for $i \in \{1, 2, 3\}$ be three orthonormal vectors, and let the state feature vectors satisfy:

1. $s_1 = \Phi_1 + \delta\Phi_3 + \eta\Phi_2$  and  2. $s_2 = \Phi_2 + \delta\Phi_3 + \eta\Phi_1$

Then it follows that the optimal $\mathcal{Q}$-function is parametrized by $\theta^* = (\theta_1^*, \theta_2^*, \theta_3^*)$ where $\theta_i^* = \Phi_i$ i.e. $\mathcal{Q}_{\theta^*}(s, a) = \mathcal{Q}^*(s, a)$ for all $s$ and $a$. Thus, according to the function $\mathcal{Q}_{\theta^*}(s, a)$, for $s_1$ the best action is $a_1$, for $s_2$ the best action is $a_2$, and in all states the second-best action is $a_3$. Next we identify the optimal perturbations used in the computation of the regularizer $\mathcal{R}(\theta^*)$ for this setting.

**Proposition 3.5.** *In the MDP $\mathcal{M}$ for any $\epsilon > 0$.*

*1. For $s = s_1 : s + \dfrac{\epsilon}{\sqrt{2}}(\theta_3^* - \theta_1^*) = \arg\max \max_{\hat{s} \in \mathcal{D}_\epsilon(s)} \max_{a \neq a^*(s)} \mathcal{Q}_{\theta^*}(\hat{s}, a) - \mathcal{Q}_{\theta^*}(\hat{s}, a^*(s))$*

*2. For $s = s_2 : s + \dfrac{\epsilon}{\sqrt{2}}(\theta_3^* - \theta_2^*) = \arg\max \max_{\hat{s} \in \mathcal{D}_\epsilon(s)} \max_{a \neq a^*(s)} \mathcal{Q}_{\theta^*}(\hat{s}, a) - \mathcal{Q}_{\theta^*}(\hat{s}, a^*(s))$*

*Proof.* We will prove item 1, and item 2 will follow from an identical argument with roles of $\theta_1^*$ and $\theta_2^*$ swapped. Let $s = s_1$. Since $a^*(s) = 1$, there are two case to consider for the maximum over $a \neq a^*(s)$, either $a = 2$ or $a = 3$. In the case that $a = 2$ we have

$$\max_{\hat{s} \in \mathcal{D}_\epsilon(s)} \mathcal{Q}_{\theta^*}(\hat{s}, a) - \mathcal{Q}_{\theta^*}(\hat{s}, a^*(s)) = \max_{\hat{s} \in \mathcal{D}_\epsilon(s)} \langle \theta_2^*, \hat{s} \rangle - \langle \theta_1^*, \hat{s} \rangle. \tag{2}$$

This is the maximum in a ball of radius $\epsilon$ around $s$ of the linear function $\langle \theta_2^* - \theta_1^*, \hat{s} \rangle$. Therefore the maximum is achieved by $\hat{s} = s + \frac{\epsilon}{\sqrt{2}}(\theta_2^* - \theta_1^*)$. The corresponding maximum value is

$$\max_{\hat{s} \in \mathcal{D}_\epsilon(s)} \langle \theta_2^*, \hat{s} \rangle - \langle \theta_1^*, \hat{s} \rangle = \langle \theta_2^* - \theta_1^*, s \rangle + \epsilon \|\theta_2^* - \theta_1^*\|_2 = \eta - 1 + \epsilon\sqrt{2}. \tag{3}$$

In the case that $a = 3$ an identical argument implies that the maximum is achieved by $\hat{s} = s + \frac{\epsilon}{\sqrt{2}}(\theta_3^* - \theta_1^*)$, with corresponding maximum value

$$\max_{\hat{s} \in \mathcal{D}_\epsilon(s)} \langle \theta_3^*, \hat{s} \rangle - \langle \theta_1^*, \hat{s} \rangle = \langle \theta_3^* - \theta_1^*, s \rangle + \epsilon \|\theta_3^* - \theta_1^*\|_2 = \delta - 1 + \epsilon\sqrt{2}. \tag{4}$$

Because $\delta > \eta$ we conclude that the value achieved in 4 is larger than that in 3. Thus the maximizer is $\hat{s} = s + \frac{\epsilon}{\sqrt{2}}(\theta_3^* - \theta_1^*)$ as desired. $\square$

In words, the optimal direction to perturb the state $s_1$ in order to have $a^*(s) \neq a^*(\hat{s})$ is toward $\theta_3^* - \theta_1^*$. Similarly for the state $s_2$, the optimal perturbation is toward $\theta_3^* - \theta_2^*$. Next we use this fact to show that in order to decrease the regularizer it is sufficient to simply increase the magnitude of $\theta_1$ and $\theta_2$, and decrease the magnitude of $\theta_3$.

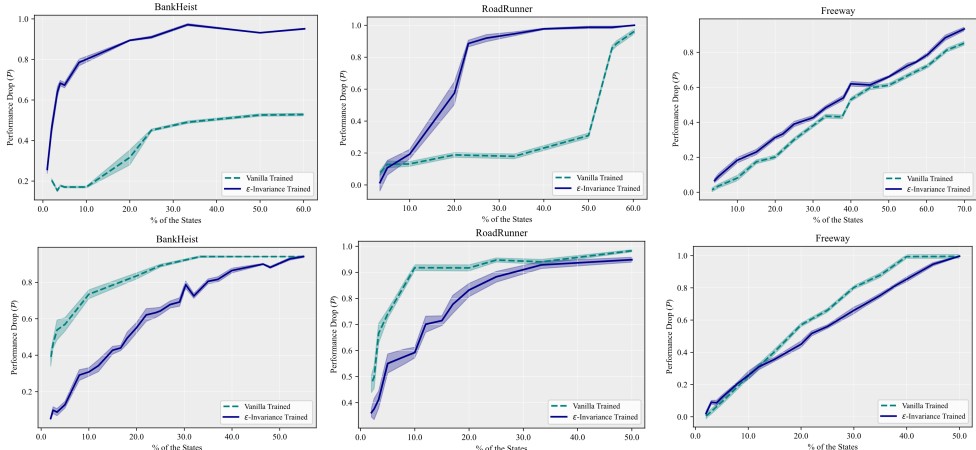

Figure 2: Up: Performance drop $\mathcal{P}_2(\Omega)$ with respect to action modification $a_2$ for the $\epsilon$-invariance and vanilla trained deep reinforcement learning policies. Down: Performance drop $\mathcal{P}_w(\Omega)$ with respect to action modification $a_w$. Left: BankHeist. Center: RoadRunner. Right: Freeway.

**Proposition 3.6.** *In the MDP $\mathcal{M}$ let $\lambda > 0$, $1/\sqrt{2} > \epsilon > 0$, and suppose that $(1-\lambda)\delta < (1+\lambda)\eta < \delta$. Let $\theta = (\theta_1, \theta_2, \theta_3)$ be given by $\theta_1 = (1+\lambda)\theta_1^*$, $\theta_2 = (1+\lambda)\theta_2^*$ and $\theta_3 = (1-\lambda)\theta_3^*$. Then $\mathcal{R}(\theta) < \mathcal{R}(\theta^*)$.*

The proof of Proposition 3.6 is provided in the supplementary material. Combining Proposition 3.6 and Proposition 3.5 we can prove the main result of this section on the effects $\epsilon$-invariance training.

**Theorem 3.7** (*Existence of Overestimation, Misalignment and Losing Counterfactuality*). *There is an MDP with linearly parameterized state-action values, optimal state-action value parameters $\theta^*$, and a parameter vector $\theta$ such that: $\mathcal{L}(\theta) < \mathcal{L}(\theta^*)$, and the parameter vector $\theta$ overestimates the optimal state-action value and re-orders the sub-optimal ones.*

The proof of Theorem 3.7 is provided in the supplementary material. The results reported in Section 4 verify the fundamental trade-off and the theoretical predictions of Section 3. In particular, across a diverse portfolio of state-of-the-art $\epsilon$-invariance training techniques that aim to obtain safe and reliable reinforcement learning, our results demonstrate that certified $\epsilon$-invariance trained policies learn misaligned, inaccurate and inconsistent values while further losing counterfactuality compared to standard reinforcement learning.

## 4 EMPIRICAL ANALYSIS IN HIGH-DIMENSIONAL MDPS

The empirical analysis is conducted in high dimensional state representation MDPs of the Arcade Learning Environment (ALE). The standard reinforcement learning policy is trained via DDQN (Wang et al., 2016) initially proposed in (van Hasselt, 2010) with prioritized experience replay proposed by (Schaul et al., 2016), and the $\epsilon$-invariance reinforcement learning policies are trained via SA-MDP RL (State Adversarial MDP, see Section 2), RADIAL (Robust Adversarial Loss-RL) (Oikarinen et al., 2021), and Optimal Robust Policy (ORP) (Li et al., 2024) where all of these influential studies were recognized with oral and spotlight presentations at NeurIPS and ICML respectively. The standard error of the mean is included for all of the figures and tables. Performance drop $\mathcal{P}$ is given by $\mathcal{P} = (\text{Score}_{\text{base}} - \text{Score}_{\text{actmod}})/(\text{Score}_{\text{base}} - \text{Score}_{\text{min}})$, where $\text{Score}_{\text{base}}$ represent the baseline run of the game without modification, $\text{Score}_{\text{min}}$ represents the minimum score available for a given game, and $\text{Score}_{\text{actmod}}$ represents the run of the game where the actions of the agent are modified for a fraction of the state observations. To measure the accuracy for the state-action value estimates formally, let $a_i$ be the $i^{\text{th}}$ best action decided by the deep neural policy in a given state $s$ (i.e. $\mathcal{Q}(s, a)$ is sorted in decreasing order, and $a_i$ is the action corresponding to $i^{\text{th}}$ largest $\mathcal{Q}$-value). For a trained agent, the value of $\mathcal{Q}(s, a_i)$ should represent the expected cumulative rewards obtained by taking action $a_i$ in state $s$, and then taking the highest $\mathcal{Q}$-value action (i.e. $a_1$) in every subsequent state. Thus, a natural test to perform would be: for a random state $s$ the policy should take action $a_i$ in state $s$, and the highest $\mathcal{Q}$-value action for the rest of the states. By comparing the relative performance drop $\mathcal{P}$ in this test to a clean run where the agent always takes the highest $\mathcal{Q}$-value action, one can measure the decline in rewards caused by taking action $a_i$. Further, we can provide a measure of accuracy for the state-action value function by comparing the results of the test for

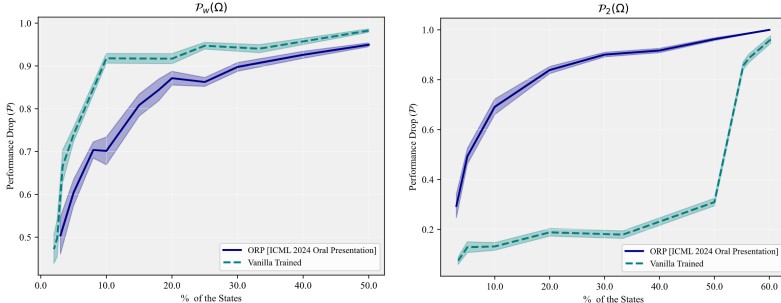

Figure 3: Loss of counterfactuality: $\mathcal{P}_2(\Omega)$ and $\mathcal{P}_w(\Omega)$ results with respect to $a_2$ and $a_w$ for ORP (Optimal Robust Policy) $\epsilon$-invariance reinforcement learning and vanilla reinforcement learning.

each $i \in \{1, 2 \dots |A|\}$, and checking that the relative performance drops $\mathcal{P}_i$ are in the correct order i.e. $0 = \mathcal{P}_1 \leq \mathcal{P}_2 \cdots \leq \mathcal{P}_{|A|}$. We take this one step further and analyze the performance drop with $\Omega$-fraction of the states in the episode uniformly at random, and making the policy execute action $a_i$ in each of the sampled states. We then record the relative performance drop as a function of $\Omega$, yielding a performance drop curve $\mathcal{P}_i(\Omega)$. More formally, the performance curve is

**Definition 4.1** (*Performance Drop Curve*). Let $\mathcal{M}$ be an MDP and $\mathcal{Q}(s, a)$ be a state-action value function for $\mathcal{M}$. In each state label the actions $a_1, \dots a_{|A|}$ in order so that $\mathcal{Q}(s, a_1) \geq \mathcal{Q}(s, a_2) \cdots \geq \mathcal{Q}(s, a_{|A|})$. The *performance drop curve* $\mathcal{P}_i(\Omega)$ is the expected performance drop of an agent in $\mathcal{M}$ which takes action $a_i$ in a randomly sampled $\Omega$-fraction of states, and executes $a_1$ in all other states.

Using these performance drop curves one can confirm whether $\mathcal{P}_i(\Omega)$ lies above $\mathcal{P}_j(\Omega)$ whenever $i > j$. Yet to be precise we will quantify the relative ordering of the performance drop curves.

**Definition 4.2** ($\tau$-*domination*). Let $\mathcal{F} : [0, 1] \to [0, 1]$ and $\mathcal{G} : [0, 1] \to [0, 1]$. For any $\tau > 0$, we say that $\mathcal{F}$ $\tau$-dominates $\mathcal{G}$ if $\int_0^1 (\mathcal{F}(\Omega) - \mathcal{G}(\Omega)) \, d\Omega > \tau$.

To compare the accuracy of state-action values for vanilla versus $\epsilon$-invariance trained agents, we can thus perform the above test, and check the relative ordering of the curves $\mathcal{P}_i(\Omega)$ using Definition 4.2 for each agent type. In addition, we can also directly compare for each $i$ the curve $\mathcal{P}_i^{\epsilon\text{-inv}}(\Omega)$ for the $\epsilon$-invariance trained agent with the curve $\mathcal{P}_i^{\text{vanilla}}(\Omega)$ of the vanilla trained agent. This is possible because $\mathcal{P}_i(\Omega)$ measures the performance drop of the agent relative to a clean run, and thus always takes values on a normalized scale from 0 to 1. Hence, an observation of $\mathcal{P}_2^{\epsilon\text{-inv}}(\Omega)$ $\tau$-dominating $\mathcal{P}_2^{\text{vanilla}}(\Omega)$ for some $\tau > 0$, would lead to the conclusion that the state-action value function of the vanilla trained agent can accurately represent the counterfactual actions while the $\epsilon$-invariance trained agents cannot.

### 4.1 LOSING INHERENT COUNTERFACTUALITY

In Section 3 we provided theoretical analysis on how $\epsilon$-invariance training affects the core principles of reinforcement learning. In this section, we demonstrate that reinforcement learning is inherently counterfactual and certified $\epsilon$-invariance training causes the policy to lose counterfactuality. Figure 2 and Figure 3 report the performance drop $\mathcal{P}_2(\Omega)$ and $\mathcal{P}_w(\Omega)$ as a function of the fraction of states $\Omega$ in which the action modification is applied for $\epsilon$-invariance and vanilla trained reinforcement learning policies. In particular, the action modification is set for the second best action $a_2$ decided by the state-action value function $\mathcal{Q}(s, a)$. As the fraction of states for $\mathcal{P}_2(\Omega)$ increases, vanilla trained deep neural policies experience lower performance drops compared to $\epsilon$-invariance. Especially in BankHeist we observe that the performance drop does not exceed $0.55$ even when the action modification is applied for a large fraction of the visited states for the standard reinforcement learning policies. This gap in the performance drop between the $\epsilon$-invariance and vanilla trained deep neural policies indicates that the state-action value function learnt by standard reinforcement learning has a better estimate for the state-action values. We further investigate the effects of $a_w = \arg\min_a \mathcal{Q}(s, a)$, i.e. worst possible action in a given state, modification on the deep neural policy. Intriguingly, Figure 2 and 3 report that the performance drop $\mathcal{P}_w(\Omega)$ is higher in the vanilla trained deep neural policies compared to $\epsilon$-invariance trained ones when the action modification is set to $a_w$. This again further verifies the theoretical predictions in Section 3 and demonstrates that standard reinforcement learning learns

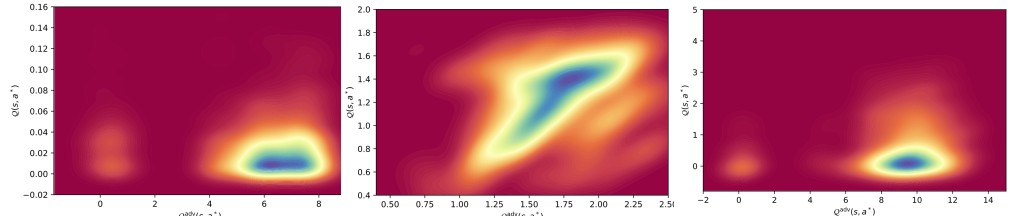

Figure 4: $\mathcal{Q}$ values of $\arg\max_{a\in\mathcal{A}}\mathcal{Q}(s,a)$ for $\epsilon$-invariance and vanilla trained deep neural policies.

| MDPs | BankHeist | | RoadRunner | | Freeway | |
|---|---|---|---|---|---|---|
| Method | $\epsilon$-Invariance | Vanilla | $\epsilon$-Invariance | Vanilla | $\epsilon$-Invariance | Vanilla |
| AM $a_2$ | 0.449±0.007 | **0.191±0.04** | 0.414±0.015 | **0.247±0.009** | 0.351±0.009 | **0.302±0.007** |
| AM $a_w$ | 0.311± 0.011 | **0.398±0.011** | 0.345±0.011 | **0.393±0.002** | 0.241±0.007 | **0.311±0.010** |

Table 1: Area under the curve of performance drop under action modification (AM) $a_2$ and $a_w$ for the state-of-the-art $\epsilon$-invariance trained deep neural policies and vanilla trained deep neural policies.

Figure 5: $\mathcal{P}_2$ and $\mathcal{P}_w$ of certified $\epsilon$-invariance training.

counterfactual and accurate state-action value functions while $\epsilon$-invariance policies lose core inherent skills of reinforcement learning.

*Reinforcement learning has inherent counterfactual ability and intrinsically learns aligned values.*

The progression of AI, from foundational work in perception to advanced decision-making, has been marked by key milestones driven by concepts drawn from biological inspiration (Treisman & Gelade, 1980; Snowden et al., 1991; Rao & Ballard, 1999; Parthasarathy et al., 2024; Newell, 1992; Imaizumi et al., 2022; Hassabis et al., 2017) Reinforcement learning is founded on the inspiration drawn from natural intelligence (ichi Amari & Arbib, 1982; Kehoe et al., 1987; Romo & Schultz, 1990; Montague et al., 1996) providing further theoretical guarantees on its limitations and capabilities (Watkins & Dayan, 1992; Barto et al., 1995). Our results show that $\epsilon$-invariance training compromises the foundational intuition of reinforcement learning, leading to a loss of the inherent counterfactuality and creating significant value misalignment. Our analysis and results demonstrate that an extensive recent line of work myopically focusing on safety in fact diverts the main contributions and the tight core connection of reinforcement learning with neuroscience while producing policies that are both in fact unsafe and misaligned. In particular, Figure 5 demonstrates that choosing the worst action leads to a smaller performance drop than choosing the second best action i.e. $\mathcal{P}_w(\Omega) < \mathcal{P}_2(\Omega)$ for all $\Omega$. Notably, these results reveal that $\epsilon$-invariance training methods assign random values to the counterfactual actions which is a direct misalignment with natural intelligence decision making. The results reported in Figure 2 demonstrate the clear juxtaposition between standard reinforcement learning and reliability-concerned reinforcement learning, i.e. $\epsilon$-invariance. Intriguingly, these findings reveal that standard reinforcement learning successfully learns aligned values and possesses an inherent capacity for counterfactual reasoning. Imposing reinforcement learning to be $\epsilon$-invariant strips out these intrinsic skills. While learning inconsistent and misaligned values can cause vulnerability problems from a security point of view, our analysis further highlights the foundational loss of information in the state-action value function as a novel fundamental trade-off intrinsic to $\epsilon$-invariance training.

*Imposing $\epsilon$-invariance causes misalignment and the loss of the inherent counterfactuality of RL.*

**Biased $\mathcal{Q}$-values in $\epsilon$-invariance Trained Deep Reinforcement Learning Policies.** In this section we investigate state-action value estimates of $\epsilon$-invariance trained and vanilla trained deep reinforcement learning policies. The results demonstrate that $\epsilon$-invariance training leads to overestimation in $\mathcal{Q}$-values which verifies the theoretical analysis provided in Section 3. In particular, Figure 4 reports the overestimation bias on the state-action values learned by the $\epsilon$-invariance trained deep neural policies. Note that the fact that $\epsilon$-invariance trained policies assign higher state-action values than the vanilla trained deep reinforcement learning policies while performing similarly, i.e. obtaining similar expected cumulative rewards, reveals that the $\epsilon$-invariance training techniques, on top of the misalignment, counterfactuality and the inaccuracy issues, learn explicitly biased state-action values.

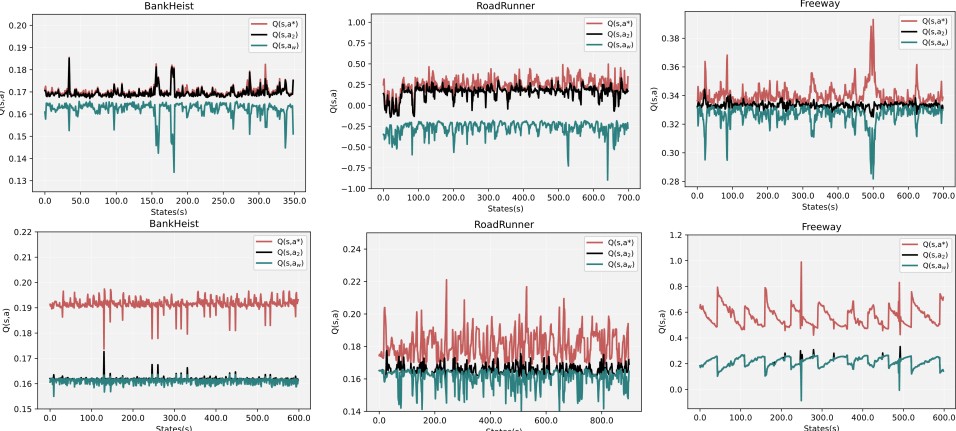

Figure 6: Normalized state-action values for the best action $a^*$, second best action $a_2$ and worst action $a_w$ over states. Up: Vanilla trained. Down: State-of-the-art $\epsilon$-local invariance trained[2].

**Action Gap Phenomenon.** The action gap is defined as the difference $\mathcal{Q}$-values, i.e. $\mathcal{G}(\mathcal{Q}, s) = \max_{\hat{a} \in \mathcal{A}} \mathcal{Q}(s, \hat{a}) - \max_{a \in \mathcal{A}_s^\perp} \mathcal{Q}(s, a)$. A connection between the action gap and the approximation errors has been discussed in prior studies and it has been hypothesized that increasing the action gap of the learned value function causes a decrease in overestimation of $\mathcal{Q}$-values. Following this study, several papers built on the hypothesis that increasing the action gap causes reduction in bias. However, our results reveal that targeting to increase the action gap must be upper-bounded by preserving the order of the counterfactual actions to obtain truly reliable and safe policies. Once this upperbound is passed the policy forms values that are misaligned and without the ability to think counterfactual. To preserve the core principles of reinforcement learning and its neuroscientific foundations that allow them to reason and generalize, we must preserve the approaches that targeted learning methods that align and match the foundational inspiration of reinforcement learning (Baird & Moore, 1993; Watkins & Dayan, 1992; Averbeck & Costa, 2017).

**A Transparent Discussion and Call for Reconsideration.** While these certified training algorithms have attracted a significant level of attention from the research community, including several spotlight and oral presentations at NeurIPS and ICML, to encourage more efforts on this line of research committing to development of responsible policies, it is more significant than ever to discuss principled investigation of these approaches. If these issues are not openly and transparently discussed, it will harm the progress towards achieving true reliability and safety while influencing future research directions and significantly pivoting research efforts. Without the principled knowledge of the actual costs and drawbacks of these algorithms a significant level of research effort might be misdirected. The results reported in Section 4.1 and Figure 5, reveal concrete problems of the $\epsilon$-invariance training techniques and how they erode reinforcement learning core skills including inherent counterfactuality and generalization. Our results reveal a fundamental trade-off between safety and alignment, and call for an urgent reconsideration of safety-concerned training of reinforcement learning and what constitutes true robustness.

## 5    CONCLUSION

In this paper, we focus on the core principles of reinforcement learning and how the inherent capabilities of RL policies are impacted by the efforts on explicit imposition of safety and robustness. We provide an extensive theoretical analysis on the fundamental effects of $\epsilon$-invariance training of reinforcement learning. Both our theoretical analysis and empirical analysis conducted in high-dimensional state representation MDPs reveal that standard reinforcement learning is inherently counterfactual and aligned with the human decision making process, while techniques focused on imposing $\epsilon$-invariance erodes core skills of reinforcement learning. Moreover, our theoretical analysis reveals that there is a fundamental trade-off, and our empirical results demonstrate $\epsilon$-invariance training breaks the core principles of reinforcement learning and causes policies to lose counterfactuality and learn misaligned and inaccurate state-action value functions. Our paper highlights transparent progress and calls for reconsideration of *robustness* and *safety*. Our analysis is critical in understanding the true capabilities of standard reinforcement learning, and opens an avenue for more principled approaches for designing algorithms to improve reliability without sacrificing the inherent essential skills of reinforcement learning that allow reasoning and generalization.

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

---

[2]Figure 6 reports that $\epsilon$-invariance training increases the action gap, yet still learns biased values. Due to the fact that the $\epsilon$-invariance trained policy has biased $\mathcal{Q}$-values, the results are reported in the normalized form in order to compare the action gaps of $\epsilon$-invariance and vanilla trained policies in the same graph.

