# Supplementary material of how to lose inherent counterfactuality in reinforcement learning

**Ezgi Korkmaz**

## 1  Algorithmic descriptions: further explanations on adversarial training

**Optimal Robust Policy (ORP) Li et al. (2024).**    Optimal Robust Policy (ORP) is a quite recently proposed adversarial training technique that is presented in ICML 2024 as an ***oral presentation***. This influential study unlike the prior studies proves that under the assumptions introduced by the paper there is in fact an optimal Bellman operator that will achieve certified adversarial, i.e. $\epsilon$-invariance, robustness.

$$\mathcal{T}_{\mathrm{car}}(s,a) = r(s,a) + \gamma \mathbb{E}_{s' \sim P(\cdot|s,a)} \left[ \min_{\hat{s}' \in B_\epsilon(s')} Q(s', \arg\max_{a_{\hat{s}'}} Q(\hat{s}', a_{\hat{s}'})) \right]$$

$\mathcal{T}_B$ is the optimal Bellman operator while canonical reinforcement learning methods focus on minimizing the error $\|\mathcal{T}_B Q_\theta - Q_\theta\|_{\mathcal{B}'}$. The ORP paper searches for the most suitable Banach space $\mathcal{B}'$ to achieve the certified adversarial training that will ensure that the policy is robust against perturbations. To achieve this first the paper introduces the $(p, d_{\mu_0}^\pi)$-seminorm definition as

$$\|f\|_{p, d_{\mu_0}^\pi} := \left( \int_{(s,a) \in S \times A} \mid d_{\mu_0}^\pi(s,a) f(s,a) \mid^p d\mu(s,a) \right)^{1/p}$$

where $d_{\mu_0}^\pi$ is a probability density function, $1 \le p \le \infty$ and $\pi, f : S \times A \to \mathbb{R}$. Then the paper further proves that minimizing the Bellman error in the $(\infty, d_{\mu_0}^\pi)$-norm will achieve certified adversarial robustness.

$$\mathcal{L}(Q_\theta; \pi_\theta) = \|\mathcal{T}_B Q_\theta - Q_\theta\|_{\infty, d_{\mu_0}^{\pi_\theta}}$$

which is equal to

$$\sup_{(s,a) \in S \times A} d_{\mu_0}^{\pi_\theta}(s,a) \max_{\hat{s} \in \mathcal{B}_\epsilon(s)} |\mathcal{T}_B Q_\theta(\hat{s}, a) - Q_\theta(\hat{s}, a)|$$

where $\pi_\theta$ is the behaviour policy induced by $Q_\theta$.

**Our Contributions [Case of ORP].**    In the main body of our paper we have conducted experiments that investigate the ORP algorithm that we have described right above. In particular, the results reported in Section 4 of the main body of our paper demonstrate that the state-of-the-art adversarial, i.e. $\epsilon$-invariance, training techniques learn misaligned and inaccurate policies while losing the counterfactuality and the core intuition of reinforcement learning.

**State Adversarial MDP (SA-MDP) Huan et al. (2020).**    State Adversarial MDP is the first study that proposed adversarial training in deep reinforcement learning and presented at NeurIPS as a ***spotlight presentation***. The paper first introduces the State Adversarial MDP concept and proposes a certified, i.e. $\epsilon$-invariance, training technique that ensures that the reinforcement learning policy will still execute the top-ranked action in the perturbed adversarial observations. The total variation distance is given by

$$D_{TV}(\pi(s, \cdot), \pi(\hat{s}, \cdot)) = \begin{cases} 0 & \arg\max_a \pi(s,a) = \arg\max_a \pi(\hat{s}, a) \\ 1 & \text{otherwise} \end{cases} \tag{1}$$

This is achieved by adding the regularizer $\mathcal{R}(\theta)$ to the temporal difference loss during training.

$$\mathcal{R}(\theta) = \sum_s (\max_{\hat{s} \in \mathcal{D}_\epsilon(s)} \max_{a \neq \arg\max_a \mathcal{Q}_\theta(s,a)} \mathcal{Q}_\theta(\hat{s}, a) - \mathcal{Q}_\theta(\hat{s}, \arg\max_a \mathcal{Q}_\theta(s,a))).$$

**Our Contributions [Case of SA-MDP].** In the main body of our paper we have conducted experiments that investigate the SA-MDP framework that we have described right above. In particular, the results reported in Section 4 and particularly Figure 3 of the main body of our paper verify the theoretical analysis provided in the main body of our paper and demonstrate that the canonical adversarial, i.e. $\epsilon$-invariance, training techniques learn misaligned and inaccurate policies while losing the counterfactuality and the core intuition of reinforcement learning.

**Robust Adversarial Loss-RL (RADIAL) Oikarinen et al. (2021).** RADIAL is another adversarial training technique introduced in (Oikarinen et al., 2021) to achieve robustness by leveraging existing neural network robustness formal verification bounds. In particular, the RADIAL algorithm proposes to optimize

$$\mathcal{L}_{\text{adv}}(\theta, \epsilon) = \mathbb{E}_{s,a,s',r}[\sum_a Q_{\text{diff}}(s,a) \cdot \text{Overlap}(s,a,\epsilon)]$$

where $Q_{\text{diff}}$ is

$$Q_{\text{diff}}(s,a) = \max(0, Q(s,a^*) - Q(s,a))],$$

and $\text{Overlap}(s,a,\epsilon)$ is

$$\text{Overlap}(s,a,\epsilon) = \max(0, \bar{Q}(s,a,\epsilon) - \underline{Q}(s,a^*,\epsilon) + \eta)$$

Overlap represents the bounds between two actions. The RADIAL algorithm also achieves certified robustness for deep reinforcement learning policies.

**Our Contributions [Case of RADIAL].** In the main body of our paper we have conducted experiments that investigate the SA-MDP and ORP which have been described in the beginning of this section. The investigation of RADIAL is reported in Section 2 of the supplementary material. The case of RADIAL also remains consistent with the case of SA-MDP and ORP where robust training techniques results in learning misaligned and inaccurate value functions while breaking the core intuition of reinforcement learning and losing counterfactuality.

**Our results demonstrate that independent from the training techniques used, robust training methods result in losing counterfactuality, breaking the core intuition of reinforcement learning and learning misaligned and inconsistent value functions.**

## 2 ANOTHER CASE: MISALIGNMENT, INACCURACIES AND LOSING COUNTERFACTUALITY OF REINFORCEMENT LEARNING

In the main body of our paper we have reported results for **State-Adversarial MDP**, i.e. SA-MDP (Huan et al., 2020), and the state-of-the-art algorithm **Optimal Robust Policy**, i.e. ORP (Li et al., 2024), that was presented as an *oral-presentation* at ICML 2024. In particular, the results reported in the main body of our paper in Figure 3, Figure 4 and Figure 6 reveal that robust trained deep reinforcement learning policies learn misaligned and inaccurate value functions while losing the counterfactual ability of standard deep reinforcement learning when it is trained with State-Adversarial MDP (SA-MDP) (Huan et al., 2020), and Optimal Robust Policy (ORP) (Li et al., 2024).

In this section we will further provide results for RADIAL (**Robust Adversarial Loss-RL**) (Oikarinen et al., 2021). In particular, Figure 1 reports results for RADIAL. The left and center column of Figure 1 demonstrate the performance drop $\mathcal{P}_2(p)$ with respect to action modification $a_2$ for the RADIAL adversarially trained deep reinforcement learning policy proposed by Oikarinen et al. (2021) and the vanilla trained deep reinforcement learning policy. The right column of the Figure 1 demonstrates the performance drop $\mathcal{P}_w(p)$ with respect to action modification $a_w$ for the RADIAL adversarially trained deep reinforcement learning policy proposed by Oikarinen et al. (2021) and the vanilla trained deep reinforcement learning policy in RoadRunner. Again the results in Figure 1 demonstrate that

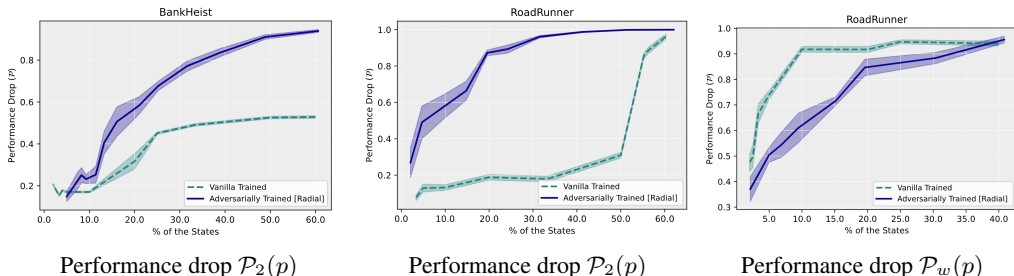

Performance drop $\mathcal{P}_2(p)$      Performance drop $\mathcal{P}_2(p)$      Performance drop $\mathcal{P}_w(p)$

Figure 1: Performance drop $\mathcal{P}_2(p)$ with respect to action modification $a_2$ for RADIAL adversarially trained deep neural policies Oikarinen et al. (2021) and vanilla trained policies. Left: BankHeist. Center: RoadRunner. Right: Performance drop $\mathcal{P}_w(p)$ with respect to action modification $a_w$ for the RADIAL adversarially trained deep neural policy and the vanilla trained deep neural policy.

the vanilla training technique has better estimates for state-action values compared to the adversarial training method RADIAL, quite recently proposed by Oikarinen et al. (2021).

In particular, the curve for $\mathcal{P}_2(p)$ for RADIAL in RoadRunner lies well above the corresponding vanilla training curve. This implies that, while taking the second best action has a relatively mild effect on the vanilla-trained policy, it causes a dramatic loss in performance for RADIAL. Similarly, the $\mathcal{P}_w(p)$ curve for RADIAL in RoadRunner lies below the corresponding curve for the vanilla-trained policy. This again implies that the vanilla-trained policy has a better estimate for which action will lead to lowest rewards than the RADIAL adversarially trained policy. The results reported in Figure 1 again demonstrate the loss of information in the state-action value function due to adversarial regulation of the temporal difference loss.

**The investigation of RADIAL and the results reported in Figure 1 once more demonstrate that $\epsilon$-invariance, i.e. adversarial, training techniques lead to learning misaligned and inaccurate value functions while breaking the core intuition of reinforcement learning and losing counterfactuality.**

We have introduced the theoretical analysis that proves the fundamental trade-off that is intrinsic to adversarial training in the main body of our paper. We have provided extensive experiments with several adversarial training methods that reveal this intrinsic fundamental trade-off in the main body of our paper. Here we report results with additional adversarial training methods and the results reveal that, independent from the training technique, robust training causes policies to learn inaccurate, misaligned and inconsistent value functions while losing counterfactuality, while standard reinforcement learning has inherent counterfactual ability.

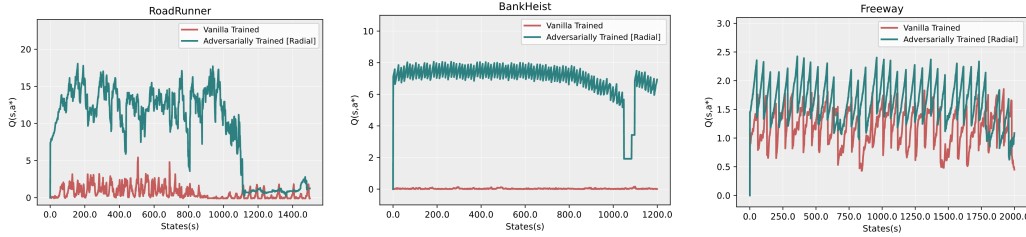

Figure 2: $Q$-value of the best action $a^*$ over the states for the RADIAL adversarially trained deep neural policy proposed by Oikarinen et al. (2021) and vanilla trained deep neural policy.

Further Figure 2 demonstrates that the overestimation bias discussed in the main body of our paper is again an issue for RADIAL as well which is a newer and different adversarial training technique as we have described above. Furthermore, exactly as the previous adversarial training methods, RADIAL also learns inaccurate, inconsistent and overestimated state-action value functions. Hence, these results once more demonstrate the loss of information in the state-action value function as a novel fundamental trade-off intrinsic to safety-concerned training in reinforcement learning.

# 3 Losing Counterfactuality: what does it entail to learn misaligned, inaccurate and inconsonant values?

Our work focuses on a line of work that claimed to obtain reliable AI systems. Our paper first introduces a theoretically well-founded analysis of these AI systems, and demonstrates that both theoretically and empirically the claimed to be reliable AI systems has issues regarding safety that has never been discussed before (Korkmaz, 2026). We further provide the connection between neural correlates of action values in natural intelligence and understanding deep neural policy decision making. Our analysis reveals that reinforcement learning is inherently aligned and has inherent counterfactual ability, while safety-concerned training methods cause standard reinforcement learning to lose counterfactuality and result in learning policies that are inaccurate, inconsistent and misaligned with human decision making processes, in which natural intelligence develops a better than random perception of counterfactual actions just like standard reinforcement learning does. Our paper provides a foundational analysis on reliability of AI systems and regarding societal impacts, our work aims to provide guidance regarding AI safety as the regulatory acts entering real-life: UK Parliament (2024); EU Parliament (2023); The White House (2023; 2024, October).

The fact that our paper explicitly theoretically and empirically reveals a fundamental trade-off between safety and alignment, and demonstrates that certified adversarially trained deep reinforcement learning policies learn inconsonant and inaccurate state-action values further implies significant concerns on the alignment with human decisions. This has been explained in Section 4. Note that humans conceptualize the values of counterfactual actions, unlike the adversarially trained deep reinforcement learning policies. See more on the human cognitive decision making process and how humans can conceptualize the set of counterfactual actions better than random here (Wunderlich et al., 2009; Hoeck et al., 2015). The results reported in Section 5 of the main body of our paper and the Section 2 of the supplementary material reveal that standard reinforcement learning is aligned with human cognitive decision making process. In fact, our results reveal that **standard reinforcement learning have inherent counterfactual ability**, and safety-concerned training techniques, as we have explained above and in the main body of our paper, result in losing counterfactual ability while producing misaligned and inconsistent value functions.

Also further note that, as also initially described in the main body of our paper in Section 2.3, recent work demonstrated vulnerabilities of certified robust, i.e. $\epsilon$-invariance, reinforcement learning policies from black-box adversarial attacks (Korkmaz, 2022; 2024) to natural attacks that revealed the generalization problems of adversarially trained deep reinforcement learning policies when compared to standard reinforcement learning (Korkmaz, 2023). While these studies highlight the safety and security problems in robust reinforcement learning policies, our paper dives into and explains the particular reasons why adversarial training experiences these safety problems. We believe it is crucial to understand the root causes of these problems regarding AI safety, because releasing models with guaranteed safety certifications with undiscovered non-robustness and vulnerabilities will in fact have serious consequences in the real world. These issues should be openly and transparently analyzed and discussed before these crucial consequences are faced in practice in real life.

# 4 Societal Impacts: AI safety requires clear and open discussion on safety premises

Our paper discovers that the promises made in *certified-safety* in fact do not hold and furthermore we lay-out the theoretical foundations on why these promises made by *certified-safety* cannot hold. We believe it is critical to openly study the exact issues arising and causing failures of machine learning systems both theoretically and empirically. Our paper discovers layers of detrimental issues with certified robust techniques. Our paper not only identifies these issues but further provides theoretical insights in to the fundamental trade-off between robustness and accuracy of the state-action value function.

# 5 Proofs of Theoretical Analysis

Below see the complete proof of Proposition 3.6. from the main body of the paper.

**Proposition 5.1.** *In the MDP $\mathcal{M}$ let $\lambda > 0$, $1/\sqrt{2} > \epsilon > 0$, and suppose that $(1-\lambda)\delta < (1+\lambda)\eta < \delta$. Let $\theta = (\theta_1, \theta_2, \theta_3)$ be given by $\theta_1 = (1+\lambda)\theta_1^*$, $\theta_2 = (1+\lambda)\theta_2^*$ and $\theta_3 = (1-\lambda)\theta_3^*$. Then $\mathcal{R}(\theta) < \mathcal{R}(\theta^*)$.*

*Proof.* By an identical argument to that in Proposition 3.5. we have that $a_2$ is always the action maximizing $\max_{a \neq a^*(s)} \mathcal{Q}_\theta(\bar{s}, a) - \mathcal{Q}_\theta(\bar{s}, a^*(s))$ whenever $(1-\lambda)\delta < (1+\lambda)\eta$. This condition is satisfied by assumption. Therefore, we conclude that for $s = s_1$, the optimal $\bar{s} \in D_\epsilon(s)$ for the scaled parameters $\theta$ is given by $\bar{s} = s + \frac{\epsilon}{\sqrt{2}(1+\lambda)}(\theta_2 - \theta_1)$. Therefore, the contribution to the sum defining $\mathcal{R}(\theta)$ from state $s_1$ is given by

$$
\begin{aligned}
\langle (\theta_2 - \theta_1), \bar{s} \rangle &= \langle (\theta_2 - \theta_1), s \rangle + \epsilon\sqrt{2}(1+\lambda) \\
&= -(1+\lambda) + (1+\lambda)\eta + \epsilon\sqrt{2}(1+\lambda)
\end{aligned}
$$

where the last step uses the fact that $s = \theta_1^* + \delta\theta_3^* + \eta\theta_2^*$ and that the vectors $\theta_i^*$ are orthonormal. Next using the fact that $(1+\lambda)\eta < \delta$ by assumption we conclude

$$
\begin{aligned}
\langle (\theta_2 - \theta_1), \bar{s} \rangle &< -(1+\lambda) + (1+\lambda)\eta + \epsilon\sqrt{2} + \epsilon\lambda\sqrt{2} \\
&< -1 + \delta + \epsilon\sqrt{2}.
\end{aligned} \tag{2}
$$

The final inequality follows from the fact that $\epsilon < \frac{1}{\sqrt{2}}$ so $\epsilon\lambda\sqrt{2} - \lambda < 0$. Switching from state $s_1$ to state $s_2$, an identical proof (with $\theta_1$ replaced by $\theta_2$) yields the same value for the contribution of state $s_2$ to the sum. By Proposition 3.5., the contribution of each type of state to the sum defining $\mathcal{R}(\theta^*)$ is

$$
\langle (\theta_3^* - \theta_1^*), s + \frac{\epsilon}{\sqrt{2}}(\theta_3^* - \theta_1^*) \rangle = -1 + \delta + \epsilon\sqrt{2}. \tag{3}
$$

Clearly the contribution of each state in 2 is strictly less than that in 3. Therefore $\mathcal{R}(\theta) < \mathcal{R}(\theta^*)$. $\quad\square$

**Theorem 5.2** (*Existence of Overestimation, Misalignment and Losing Counterfactuality*)**.** *There is an MDP with linearly parameterized state-action values, optimal state-action value parameters $\theta^*$, and a parameter vector $\theta$ such that: $\mathcal{L}(\theta) < \mathcal{L}(\theta^*)$, and the parameter vector $\theta$ overestimates the optimal state-action value and re-orders the sub-optimal ones.*

*Proof.* Let $\mathcal{M}$ be the MDP in the setting of Proposition 3.5 and define $\theta$ as in Proposition 3.6 by setting $\theta_1 = (1+\lambda)\theta_1^*$, $\theta_2 = (1+\lambda)\theta_2^*$, and $\theta_3 = (1-\lambda)\theta_3^*$. The overall regularized loss has the form $\mathcal{L}(\theta) = \mathcal{TD}(\theta) + \mathcal{R}(\theta)$. Where $\mathcal{TD}(\theta)$ is the standard temporal difference loss. For the MDP $M$ and parameters $\theta$ we can explicitly calculate this loss:

$$
\begin{aligned}
\mathcal{TD}(\theta) &= \frac{1}{6} \sum_{i=1}^{2} \sum_{j=1}^{3} (r(s_i, a_j) + \gamma \max_k \langle \theta_k, s_{3-i} \rangle - \langle \theta_j, s_i \rangle)^2 \\
&\leq \frac{1}{6} \sum_{i=1}^{2} \sum_{j=1}^{3} (r(s_i, a_j) + \gamma \max_k (1+\lambda)\langle \theta_k^*, s_{3-i} \rangle - (1-\lambda)\langle \theta_j^*, s_i \rangle)^2 \\
&= \frac{1}{6} \sum_{i=1}^{2} \sum_{j=1}^{3} (r(s_i, a_j) + \gamma \max_k \langle \theta_k^*, s_{3-i} \rangle - \langle \theta_j^*, s_i \rangle + \lambda\gamma \max_k \langle \theta_k^*, s_{3-i} \rangle + \lambda\langle \theta_j^*, s_i \rangle)^2 \\
&= \frac{1}{6} \sum_{i=1}^{2} \sum_{j=1}^{3} (\lambda\gamma \max_k \langle \theta_k^*, s_{3-i} \rangle + \lambda\langle \theta_j^*, s_i \rangle)^2
\end{aligned}
$$

where the final equality follows from the optimality of the paramters $\theta^*$. Using the fact that $\langle \theta_j^*, s_i \rangle \leq 1$ for all $i, j$ we conclude that $\mathcal{TD}(\theta) \leq (\gamma\lambda + \lambda)^2 < 4\lambda^2$. Thus, for $\lambda < \frac{1}{4}$ we have by Proposition 3.6 $\mathcal{TD}(\theta) \leq 4\lambda^2 < \lambda < \mathcal{R}(\theta^*) - \mathcal{R}(\theta)$. Therefore $\mathcal{L}(\theta) < \mathcal{L}(\theta^*)$. Clearly, $\theta$ overestimates the optimal state-action values in both $s_1$ and $s_2$ by a factor of $1 + \lambda$. Furthermore, setting $\lambda$ such that $\frac{1+\lambda}{1-\lambda} > \frac{\delta}{\eta}$ implies that $a_3$ will be the third ranked action in both states $s_1$ and $s_2$ i.e. that $\theta$ leads to re-ordering of the suboptimal actions. $\quad\square$