# OpenReview forum: "How to Lose Inherent Counterfactuality in Reinforcement Learning"
_ICLR.cc/2026/Conference — ICLR 2026 Poster_

### Official Review · Reviewer_LRJj · 2025-10-30

**Soundness:** 4
**Presentation:** 3
**Contribution:** 4
**Rating:** 8
**Confidence:** 5

**Summary:**

This paper provides empirical and theoretical evidence challenging the prevalence of $\epsilon$-invariant adversarial retraining in robust reinforcement learning (RL), and provides a new dimension of analysis to the performance-robustness tradeoff in adversarial RL. Specifically, the paper provides proofs that policies can either be $\epsilon$-invariant or optimally estimate state-action values, but not both. The empirical results follow this conclusion, showing that when $\epsilon$-invariant policies are forced to choose actions based on counterfactual observations, performance degradation is worse than would otherwise be seen with nominal policies.

**Strengths:**

- The paper provides strong theoretical and intuitive results showcasing the downsides of commonly used robust RL frameworks.
- The work is very insightful and gives strong justifications for where the field of adversarial robust RL should be looking.

**Weaknesses:**

#### Preliminaries
By reading the paper, the term _counterfactuality_ can be implicitly understood as the accuracy of the value function for non-optimal actions. The paper would benefit from an explicit definition, as it would help understand the implications of losing counterfactuality.

#### Minor formatting:
- Figure 6 caption, "Lipschitz" has a "c".
- In section 2, concerns on reliability, the citation for [1] has the author's first and last name switched, should be Zhang et. al. (dblp: https://dblp.org/rec/conf/nips/0001CX0LBH20.html)

[1] Huan Zhang, Hongge Chen, Chaowei Xiao, Bo Li, Mingyan Liu, Duane S. Boning, Cho-Jui Hsieh: Robust Deep Reinforcement Learning against Adversarial Perturbations on State Observations. NeurIPS 2020

**Questions:**

**Questions**
- Is counterfactuality well-defined?
- Why is counterfactuality important in Q functions, beyond conceptual links to neuroscience? Policy gradient methods, such as PPO, do not use the Q/Value function after training, so it would seem that counterfactuality is not meaningfully leveraged for its generalizability.
- Some more recent robust RL methods optimize other objectives, such as maximin value [2] and minimum regret [3], instead of value-optimal invariant actions. Can the conclusions in this paper be extended to have implications for those methods as well?
- Theorem 3.3 proves the inherent tradeoff between accurate Q functions and robustness. If $\epsilon$-invariance is enforced in the policy objective only, is this a potential solution to the described problem?
- Also in Theorem 3.3, it is stated that the two functions $Q^*$ and $Q_\theta$ 'agree' on certain states. Is the agreement an equivalence of value, or is it referring to the ordinal rank of actions?

Lastly, a comment: Section 4.1 describes that the performance drop of vanilla policies is larger than that of robust policies for the worst-case action. It could be argued that the robust methods are succeeding in their objective here.

[2] Yongyuan Liang, Yanchao Sun, Ruijie Zheng, Furong Huang: Efficient Adversarial Training without Attacking: Worst-Case-Aware Robust Reinforcement Learning. NeurIPS 2022

[3] Roman Belaire, Arunesh Sinha, Pradeep Varakantham: On Minimizing Adversarial Counterfactual Error in Adversarial Reinforcement Learning. ICLR 2025

---

> ### Author Response · Authors · 2025-11-22
>
> We are truly glad to hear that you found that our work provides strong justifications with very insightful, strong theoretical and intuitive results introducing a new dimension of analysis. We are truly glad to see your recognition of our work with regard to soundness and contributions as excellent. We highly appreciate your positive and valuable review.
>
> 1. *“By reading the paper, the term counterfactuality can be implicitly understood. Is counterfactuality well-defined? The paper would benefit from an explicit definition.”*
>
> Thank you very much for this insightful comment. We can definitely add a more explicit and formal definition of counterfactuality.
>
> 2. *”Why is counterfactuality important in Q functions, beyond conceptual links to neuroscience? Policy gradient methods, such as PPO, do not use the Q/Value function after training, so it would seem that counterfactuality is not meaningfully leveraged for its generalizability.”*
>
> In policy gradient methods, the policy and $\mathcal{Q}$-function are fundamentally intertwined. In policy gradient methods like PPO, the main policy optimization objective is guided by the value function where the policy puts higher probability mass on higher value actions, and lower mass on lower value actions. Hence, as in the case of $\mathcal{Q}$-learning, the agent learns a policy with lack of counterfactuality as it is essentially guided by the $\mathcal{Q}$/value function during training.
>
> 3. *”Some more recent robust RL methods optimize other objectives, such as maximin value [2] and minimum regret [3], instead of value-optimal invariant actions. Can the conclusions in this paper be extended to have implications for those methods as well?”*
>
> The study [2] uses the same measure of robustness inside the regularizer and weighs the significance of achieving robustness in specific states. Thus, incorporating the same measure of robustness will critically also be prone to exhibit the similar behaviour described in our paper. From a theoretical perspective, given the results of [4] with regard to $\epsilon$-local invariance in high-dimensional spaces, one can expect that the optimal solution to the objective of the mentioned work still learns an $\epsilon$-local invariant policy when given the objective function based on minimum regret.
>
> [1] Robust Deep Reinforcement Learning against Adversarial Perturbations on State Observations, NeurIPS 2020.
>
> [2] Efficient Adversarial Training without Attacking: Worst-Case-Aware Robust Reinforcement Learning. NeurIPS 2022.
>
> [3] On Minimizing Adversarial Counterfactual Error in Adversarial Reinforcement Learning, ICLR 2025.
>
> [4] Towards Optimal Adversarial Robust Q-learning with Bellman Infinity-error, ICML 2024.
>
> 4. *”Theorem 3.3 proves the inherent tradeoff between accurate Q functions and robustness. If  $\epsilon$-invariance is enforced in the policy objective only, is this a potential solution to the described problem?”*
>
> Thank you for this insightful question. As also related to right above, in policy gradient methods, the policy and $\mathcal{Q}$-function are fundamentally intertwined. The $\mathcal{Q}$/value function update in policy gradient methods uses the current policy to estimate the value function. Thus, when $\epsilon$-invariance is enforced on the policy, this directly affects the value estimates of the policy and hence the value function. Since these value estimates are used to optimize the policy, this will directly in fact affect the policy.
>
> 5. *”Also in Theorem 3.3, it is stated that the two functions $\mathcal{Q}^{\*}$ and $\mathcal{Q}^\theta$  'agree' on certain states. Is the agreement an equivalence of value, or is it referring to the ordinal rank of actions?”*
>
> In the proof of Theorem 3.3, agreement refers to equivalence of value.
>
> We highly appreciate again that our efforts to produce an intuitive and rigorous analysis were recognized by you.

---

### Official Review · Reviewer_WJm3 · 2025-10-31

**Soundness:** 2
**Presentation:** 3
**Contribution:** 3
**Rating:** 4
**Confidence:** 3

**Summary:**

The paper claims that $\varepsilon$-locally invariant ($\varepsilon$-LI) training—defined as keeping the optimal action unchanged for any perturbed state $\bar{s}$ inside an $\varepsilon$-ball—breaks RL’s inherent counterfactuality and yields misaligned or over-smoothed $Q$-functions.
However, its reading of prior “observation-robust” RL (notably Huan et al., NeurIPS 2020) appears stronger than what that work actually proves.
Huan formalizes a state-adversarial MDP with a perturbation set $B(s)$, analyzes value functions under an optimal adversary, and introduces regularizers or certificates that bound changes, not strict invariance of $\arg\max_a Q(s,a)$ for all $\bar{s}\in B(s)$.
By building its theory on strict $\varepsilon$-invariance (argmax stability over the entire $\varepsilon$-ball), the paper’s counterfactuality-loss claims do not directly follow from Huan’s bounded-difference / certification framework.

**Strengths:**

- Raises a timely and interesting question: Can robustness training erase counterfactual structure?
- Clear toy-MDP reasoning (e.g., linear function arguments around $\varepsilon$-balls) clarifies a plausible accuracy–invariance tension.
- Provides a useful critique that encourages more precise definitions of “robustness” targets in RL (invariance vs bounded sensitivity).

**Weaknesses:**

- Over-interpreting prior work as $\varepsilon$-invariance.

The paper states that $\varepsilon$-LI methods “form the foundations of robust RL” and implies that Huan et al. (2020) learn $\varepsilon$-invariant policies.
However, Huan formalizes $B(s)$, analyzes SA-MDP values under an optimal adversary, and proposes regularizers/certificates that bound changes —not a guarantee that $\arg\max_a Q(s,a)$ is identical for all $\bar{s}\in B(s)$.

- Assumption-driven logical dependency.

The paper’s core theoretical framework (Def. 3.2—$\varepsilon$-LI and subsequent theorems) assumes strong argmax invariance.
Huan’s mathematics, however, deals with continuity/boundedness and partial certification (fractional guarantees).
Therefore, the paper’s counterfactuality-loss conclusion, which depends on full invariance, cannot be directly derived from the premises.

- Ambiguity in generalization scope.

The paper mainly targets the observation-robust line of research but does not rigorously distinguish whether the same “counterfactuality loss” applies to action-robust, transition-robust, or reward-robust settings.
Since each robustness category defines perturbation over different MDP components, it is unclear whether the claimed phenomenon generalizes.

**Questions:**

- Definition consistency (core issue).

The paper’s Definition ($\varepsilon$-LI: argmax invariance within the $\varepsilon$-ball) and Huan et al.’s bounded-change/certification results differ in strength.
Which prior methods actually satisfy your definition of $\varepsilon$-LI?
If none, wouldn’t it be more appropriate to describe your work as a limit-case analysis—a theoretical exploration of what would happen under strong invariance—rather than as a critique of existing algorithms?

- Replacement with bounded-difference assumption.

If the invariance condition were replaced by Huan’s bounded-difference assumption—e.g.,
$|Q(s,a)-Q(\bar{s},a)|$ or $D_{TV}(\pi(\cdot|s),\pi(\cdot|\bar{s}))$ being bounded—
how would your counterfactuality-loss theorem change?
Would the same qualitative result hold, or would the strength of the conclusion weaken?

- Applicability across robustness categories.

Beyond observation-robust settings, does the proposed “loss of counterfactuality” also occur in action-robust, transition probability-robust, or reward-robust RL algorithms?
Since the definition of robustness (what is being regularized or perturbed) varies across these classes, the effect on $Q$-ordering may also differ.
Could you clarify this distinction theoretically or empirically?

- Supplementary/Appendix clarification.

Supplementary material(L229) references Appendix, which is not included in the submission.
Instead of adding it to the supplementary, it may be clearer and more appropriate to include those contents directly as a Main Paper Appendix, so that key theoretical or experimental details are accessible without external files.

---

> ### Author Response · Authors · 2025-11-22
>
> We are glad to see you found that our paper raises a timely and interesting question with clear reasoning and provides a useful critique that encourages more precise definitions of “robustness” targets in reinforcement learning and thank you for valuing the presentation and the contributions of our paper highly.
>
> 1. *”Over-interpreting prior work as $\epsilon$-invariance. Huan et al. does not a guarantee that $\text{argmax}_a \mathcal{Q}(s,a)$ is identical for all $\hat{s} \in B(s)$.”*
>
> We would like to kindly point out that the definition of Huan et al. is precisely the definition of $\epsilon$-local invariance. Please see Section 3.4 of Huan et al. [1]. Here it can be clearly seen that their optimal adversary definition is exactly $\text{argmax}_a \mathcal{Q}(s,a)$ does not change under perturbations. To directly quote Huan et al. : “The aim is to keep the top-1 action to stay unchanged after perturbation.” This is precisely the definition of $\epsilon$-local invariance.
>
> 2. *”Replacement with bounded-difference assumption.”*
>
> One can immediately verify that Theorem 3.3 in fact proves that the optimal state-action value function $\mathcal{Q}^{\*}$ is not $(\epsilon,\delta)$-invariant for any $\delta < 0.7 \cdot \epsilon$ under the bounded difference assumption where $\lvert \mathcal{Q}(s,a^{\*}) - \mathcal{Q}(s’,a^{\*})\rvert < \delta$. Furthermore, note that the results of Theorem 3.3 are even stronger for the case of bounded difference assumption where $\mathcal{Q}_\theta$ is $(\epsilon,\delta)$-locally invariant for any $\epsilon <  0.1$ and $\delta = \epsilon/10$, which is a much stronger local invariance bound.
>
> 3. *”Robustness in other MDP components”*
>
> State observation robustness is the most general framework of robustness that has been discussed in the literature extensively. This is because introducing perturbations to the state observations essentially captures disturbances to the policy, actions and transitions. Hence, our paper focuses on the most general framework of robustness in deep reinforcement learning to provide a transparent and comprehensive analysis.

---

### Official Review · Reviewer_t3Jn · 2025-11-01

**Soundness:** 2
**Presentation:** 2
**Contribution:** 2
**Rating:** 4
**Confidence:** 1

**Summary:**

This paper shows that enforcing ϵ-local invariance—a common technique to stabilize deep reinforcement learning policies—undermines the natural counterfactuality of value estimation, leading to inconsistent and misaligned policies. The authors argue this approach contradicts the biological principles underlying reinforcement learning, creating a gap between artificial and natural intelligence. Their findings urge a rethinking of regularization methods to preserve counterfactual reasoning while ensuring robustness.

**Strengths:**

I'm not an expert in this area and don't understand much about this work.

**Weaknesses:**

See Questions.

**Questions:**

- What is an action modification exactly?
- What is the definition of $\mathcal{P}_{w}$? What is $w$ in it?
- I don't understand Figure 4. What is the x-axis? What does the color represent exactly?
- What is the definition of counterfactuality under RL settings? How do you measure it exactly?

---

> ### Author Response · Authors · 2025-11-22
>
> Thank you for investing your time to provide a sincere review for our paper.
>
> 1. *"What is an action modification exactly?"*
>
> Action modification is described between Line 315 and 323 and it refers to allowing the agent to pick $a_i$ where $a_i$ is the $i^{\textrm{th}}$ best action.
>
> 2. *"What is the definition of $\mathcal{P}_w$ ? What is $w$ in it?"*
>
> $\mathcal{P}$ is the performance drop and it is formally defined in Line 312. $w$ here refers to the $a_w = \text{argmin}_a \mathcal{Q}(s,a)$ and this is also defined in Line 372.
>
> 3. *"I don't understand Figure 4. What is the x-axis? What does the color represent exactly?"*
>
> The x-axis in Figure 4 is the state-action values of $a^{\*}$ for the adversarially trained deep reinforcement learning policy and the y-axis is the state-action values of $a^{\*}$ for the standard deep reinforcement learning policy. The way to read Figure 4 is to observe where the yellow and blue density lies. These plots demonstrate the overestimation of $\mathcal{Q}$ values for the adversarially trained policies.
>
> 4. *"What is the definition of counterfactuality under RL settings? How do you measure it exactly?"*
>
> Counterfactuality is the accuracy of the value function for non-optimal actions and how it is measured is explained in Section 4 right before Section 4.1.

---

### Official Review · Reviewer_vbm4 · 2025-11-02

**Soundness:** 3
**Presentation:** 3
**Contribution:** 4
**Rating:** 8
**Confidence:** 3

**Summary:**

The authors study recently developed methods for RL training that enforce ϵ-local invariance using theoretical and empirical analysis. They show that such methods have significant problems when compared to more traditional methods.

**Strengths:**

**Clear organization and writing**: While there are some improvements to be made (see below), the overall organization and writing of the paper is clear.

**Important topic**: This topic is clearly important. Safety, reliability, trustworthiness, and robustness have been a major theme of research in RL over the past several years, and this paper is an important re-examination of some of the consequences of current solutions.

**Both theoretical and experimental evidence**: The authors provide both theoretical and more concrete and realistic empirical evidence. This should be more strongly leveraged in the paper — for example, by more clearly identifying intuitions and mechanisms (see below) — but the theory and experiments are there to do that.

**Strong results**: The results, particularly in the experimental sections, are fairly strong. It’s clear that there are major differences between the training methods that enforce ϵ-local invariance and more traditional methods. Ultimately, this is what matters.

**Weaknesses:**

**Insufficiently explained impacts**: Many of the effects of enforcing ϵ-local invariance that are noted by the authors are sufficiently tied to impacts in both the abstract and introduction. For example, in the paper’s abstract, the authors say that such training “cause policies to lose counterfactuality”, “result in learning misaligned and inconsistent values”, “break the core intuition and the true biological inspiration of reinforcement learning”, and “introduce an intrinsic gap between how natural intelligence understands and interacts with an environment in contrast to AI agents”. These sound bad, but what are the impacts on performance? That is, under what circumstances and by how much does this training approach reduce concrete performance measures such as long-term reward across some set of environments? Note that the performance impacts are made clear in the experimental section, but that should be carried through into the abstract and introduction.

**Hyperbolic language**: The authors use hyperbolic language throughout the paper. For example, in the paragraph on contributions, the authors refer to “our comprehensive study” (when “our study” would be adequate), refer to “a theoretically well-founded rigorous analysis” (when “a theoretical analysis” would be adequate), and note that “ϵ-local invariance training shatters this elegant relationship” (when “ϵ-local invariance training disrupts this relationship” would be adequate). The authors should use more neutral language throughout the paper.

**Lack of clear intuition and mechanisms**: The authors state high-level ideas in the introduction and elsewhere (e.g., “Our extensive analysis and results discover that ϵ-invariance training methods break the core intuitive principles of reinforcement learning…”), and they provide low-level theorems and experiments. However, the intuitions behind those theorems and the mechanisms to that lead to the experimental results are not made clear. The paper would be greatly improved by more mid-level intutions and mechanisms for how ϵ-local invariance affects policies (and, thus, performance).

**Overly broad conclusions**: The paper includes statements such as “Reinforcement learning has inherent counterfactual ability…” However, the paper clearly cannot provide evidence for *all* RL methods, so statements such as this should be limited to some set of methods. Additionally, the paper does not define “inherent counterfacual ability.” The conclusions should be stated more narrowly.

**Questions:**

The abstract notes that “…this line of [ϵ-local invariance] training methods break the core intuition and the true biological inspiration of reinforcement learning...” Why is this a problem? You outline many other problems, but I don’t see why these are a particular problem.

The introduction notes that “Our analysis reveals that reinforcement learning possesses an inherent ability for counterfactual reasoning and is naturally aligned with human decision-making processes…” Concretely, what does this mean?

---

> ### Author Response · Authors · 2025-11-22
>
> We highly appreciate your thorough and insightful review, and we are truly glad to hear that you found that our paper focuses on clearly an important topic that has been a major theme of research in RL and our work provides both theoretical and more concrete and realistic empirical evidence with clear and strong results. We are glad to see your recognition of the contributions of our paper and your assessment of our paper as excellent. We sincerely appreciate your valuable, in-depth and insightful review.
>
> 1. *”The abstract notes that “…this line of [ϵ-local invariance] training methods break the core intuition and the true biological inspiration of reinforcement learning...” Why is this a problem? You outline many other problems, but I don’t see why these are a particular problem.”*
>
> The foundation and the core principles of reinforcement learning were built on the biological inspiration and the mathematically formalized model of the dopaminergic system in the brain, where an agent learns through trial-and-error interacting with an environment by adjusting its expectation based on the difference between the expected reward and the actual reward. This inspiration demonstrated that a biological learning structure can be abstracted into a powerful, general-purpose computational framework which allowed us to solve one of the most complex tasks known to date. Biological systems still to this day represent generally intelligent agents that excel at adapting to novel and changing environments with remarkable efficiency and robustness. Although AI does not have to resemble or imitate biological systems, thus far this inspiration provided one of the strongest and the most effective ways of learning. The current success of reinforcement learning and its foundations carries a critical potential for creating truly general, adaptive, and efficient artificial general intelligence agents. Hence, research directions that might alter this core intuition of reinforcement learning carry a potential to deviate from this objective which in our paper we demonstrate that $\epsilon$-local invariance training methods lead to loss of essential skills including the inherent ability for counterfactual reasoning of reinforcement learning. We would be also happy to add this explanation into the paper to provide further insights.
>
>
>
> 2. *”The introduction notes that “Our analysis reveals that reinforcement learning possesses an inherent ability for counterfactual reasoning and is naturally aligned with human decision-making processes…” Concretely, what does this mean?”*
>
> Neuroscientific results reveal that natural intelligence preserves a clear ordering of both factual and counterfactual outcomes and this mechanism is a key attribute of the decision-making process that enables generalization and reasoning and serves to inform future decision-making. Our results demonstrate that standard reinforcement learning also inherently has this ability, which is naturally aligned with human decision-making processes.
>
> We sincerely thank you for all of your insightful and constructive comments. We are more than happy to incorporate them, as they will undoubtedly perfect the presentation.

---

### Author Response · Authors · 2025-12-03

Dear All,

We thank all of you for the time you have invested in providing a review for our paper. We are truly glad to see that

**Reviewer vbm4** found that our paper focuses on clearly an important topic that has been a major theme of research in reinforcement learning and our work provides both theoretical and more concrete and realistic empirical evidence with clear and strong results. We are glad to see Reviewer vbm4’s recognition of the contributions of our paper and their assessment of our paper as excellent.

**Reviewer WJm3** found that our paper raises a timely and interesting question with clear reasoning and provides a useful critique that encourages more precise definitions of “robustness” targets in reinforcement learning and we are glad to see that the reviewer valued the presentation and  the contribution of our paper highly.

**Reviewer t3Jn** found that our findings urge a rethinking of methods on regularization and robustness to preserve counterfactual reasoning while ensuring robustness.

**Reviewer LRJj** found that our work provides strong justifications with very insightful, strong theoretical and intuitive results introducing a new dimension of analysis. We are truly glad to see Reviewer LRJj’s recognition of our work with regard to soundness and contributions as excellent.

We sincerely thank all of you.

---

### Meta-Review · Area_Chair_GPvN · 2026-01-02

**Summary:**

This paper studies the consequences of enforcing ε-local invariance in reinforcement learning and argues—both theoretically and empirically—that such robustness constraints can destroy counterfactual value estimation and lead to misaligned or degraded policies. The submission received strong support from a majority of reviewers, including two expert reviewers who rated the contribution as excellent and highlighted the importance of the question, the clarity of the theoretical analysis, and the thorough experimental validation. While some reviewers raised concerns about the strength of the assumptions and the breadth of the claims, the authors provided detailed rebuttals and clarifications, and the empirical results consistently support the paper’s central message. I therefore recommend acceptance based on the overall reviewer consensus and the depth of the experimental evidence. That said, I would like to note a remaining concern regarding the theoretical justification: the ε-invariance phenomenon and the associated robustification methods largely arise in the context of non-linear function approximation over high-dimensional inputs (e.g., images), where valid states are sparse and well-separated in representation space. In such regimes, a sufficiently expressive function approximator should, in principle, be able to enforce local invariance without materially increasing approximation error for the original training objective. As a result, I am not fully convinced that the theoretical tradeoff identified in the paper captures the typical operating regime of practical robust RL methods. Nonetheless, the paper raises an important and timely perspective that is likely to stimulate valuable discussion and further research, justifying acceptance.

**Reviewer Concerns:**

Most are addressed, though I have concerns of my own.

**Reviewer Scores:**

I think the reviewers would vote for acceptance.

---

### Decision · Program_Chairs · 2026-01-26

Accept (Poster)